# Short-Term Effect of Percutaneous Coronary Intervention on Heart Rate Variability in Patients with Coronary Artery Disease

**DOI:** 10.3390/e23050540

**Published:** 2021-04-28

**Authors:** Chang Yan, Changchun Liu, Lianke Yao, Xinpei Wang, Jikuo Wang, Peng Li

**Affiliations:** 1School of Control Science and Engineering, Shandong University, Jinan 250061, China; cyan@seu.edu.cn (C.Y.); 201313018@mail.sdu.edu.cn (L.Y.); wangxinpei@sdu.edu.cn (X.W.); william2018sanguine@gmail.com (J.W.); 2Division of Sleep and Circadian Disorders, Brigham and Women’s Hospital, Harvard Medical School, Boston, MA 02115, USA

**Keywords:** area index (AI), autonomic function, complexity, heart rate asymmetry (HRA)

## Abstract

Myocardial ischemia in patients with coronary artery disease (CAD) leads to imbalanced autonomic control that increases the risk of morbidity and mortality. To systematically examine how autonomic function responds to percutaneous coronary intervention (PCI) treatment, we analyzed data of 27 CAD patients who had admitted for PCI in this pilot study. For each patient, five-minute resting electrocardiogram (ECG) signals were collected before and after the PCI procedure. The time intervals between ECG collection and PCI were both within 24 h. To assess autonomic function, normal sinus RR intervals were extracted and were analyzed quantitatively using traditional linear time- and frequency-domain measures [i.e., standard deviation of the normal-normal intervals (SDNN), the root mean square of successive differences (RMSSD), powers of low frequency (LF) and high frequency (HF) components, LF/HF] and nonlinear entropy measures [i.e., sample entropy (SampEn), distribution entropy (DistEn), and conditional entropy (CE)], as well as graphical metrics derived from Poincaré plot [i.e., Porta’s index (PI), Guzik’s index (GI), slope index (SI) and area index (AI)]. Results showed that after PCI, AI and PI decreased significantly (*p* < 0.002 and 0.015, respectively) with effect sizes of 0.88 and 0.70 as measured by Cohen’s *d* static. These changes were independent of sex. The results suggest that graphical AI and PI metrics derived from Poincaré plot of short-term ECG may be potential for sensing the beneficial effect of PCI on cardiovascular autonomic control. Further studies with bigger sample sizes are warranted to verify these observations.

## 1. Introduction

Percutaneous coronary intervention (PCI) can significantly improve myocardial ischemia in patients with coronary artery disease (CAD) [1] and can reduce the mortality in patients with unstable CAD [2]. Since the PCI procedure is relatively safe, efficient, and with a fast recovery phase, it has been increasingly used and has become the gold standard for coronary revascularization in CAD patients. However, the PCI procedure does not render a cure for CAD; the effectiveness of PCI and whether myocardial infarction reoccurs after PCI procedure need special attention. To facilitate rehabilitation after this procedure, non-invasive monitoring tools for assessing the cardiovascular system are required.

Heart rate variability (HRV), i.e., the fluctuation in the time intervals between normal sinus heart beats, is a commonly-used noninvasive tool in clinical practice for assessing the autonomic regulation, which is linked to cardiovascular function and predicts cardiovascular risk in healthy individuals [3,4,5,6]. Several studies have investigated the use of HRV for prognostic purposes in CAD patients. For example, Harris et al. [7] reported that analysis of 24 h HRV before PCI procedure in patients with acute coronary syndrome provided incremental prognostic information about the risk of rehospitalization and mortality in one year. In terms of short-term effect of the treatment on HRV. Bonnemeier et al. [8] analyzed HRV measured from 24-h Holter monitoring found that the standard deviation of HRV is increased after successful coronary reperfusion. Abrootan et al. [9] evaluated the HRV by using 5 min RR intervals by time-domain indices and observed a similar increase in the standard deviation of HRV within 24 h after PCI in patients with stable angina pectoris.

However, mainly traditional methods were used in the above-mentioned studies. The past two or three decades have witnessed the advance of novel measures derived from graphical representation of time-series, such as the Poincaré plot, or based on nonlinear dynamical theory [10,11] that are able to capture additional valuation information from HRV. Specifically, Poincaré plot derived metrics such as Porta’s index (PI) [12], Guzik’s index (GI) [13], slope index (SI) [14], and area index (AI) [15] have been used to study the asymmetry (i.e., heart rate asymmetry, HRA) to understanding the uneven contribution of decelerations and accelerations to the HRV. Among the commonly used nonlinear measures, many entropy metrics have been widely used especially for analyzing short-term HRV time-series, such as approximate entropy (ApEn) [16], sample entropy (SampEn) [17], fuzzy entropy (FuzzyEn) [18], distribution entropy (DistEn) [11], and conditional entropy (CE) [19]. In prior studies, changes of HRA in pathological conditions have been demonstrated based on these HRA indices [14,15]. The entropy measures have been used to have been used to assist the prediction of cardiovascular disease outcomes [20]. They both have been applied to remote monitor of obese children [21], automated predict sudden cardiac death [22], and study the impact of smoking on heart rate variability among middle age men [23].

There is a lack of studies that systematically examine the value of HRV to assess the autonomic response to PCI treatment by using time- and frequency- domain measures, Poincaré indices and entropy metrics. To fill this knowledge gap, we sought to first elucidate the autonomic response in short term after PCI procedure in CAD patients. To achieve this, we analyzed data of 27 CAD patients who were admitted for PCI procedure. Resting state electrocardiogram (ECG) signals were collected both before and after the procedure with both time lags to PCI procedure within 24 h. Normal sinus to normal sinus beats intervals were extracted from the ECG signals and were analyzed respectively by prior established time- and frequency-domain parameters, entropy measures and Poincaré metrics.

## 2. Materials and Methods

### 2.1. Participants

We revisited a set of 5-min ECG data collected from a previous human study we conducted during 2013–2018 [11,24,25,26,27]. CAD patients were recruited from those who were scheduled with an interventional surgery and data were collected 24-h before the surgery. For this specific study, we only included those whose coronary angiography demonstrated at least one main coronary branch with a >50% stenosis. Patients with left ventricular ejection fraction <50% (echocardiography) were excluded to eliminate possible influence of heart failure. Patients with frequent ectopic beats (routine ECG) were also excluded. Table 1 shows their basic characteristics.

### 2.2. Protocols

Measurements were taken in a quiet, temperature-controlled clinical measurement room (25 ± 3 °C) at Shandong Provincial Qianfoshan Hospital, Shandong University, by a cardiovascular function detection device (CV FD-II, Huiyironggong Technology Co., Ltd., Jinan, China). Five-minute standard limb lead II ECG data were recorded continuously at a sampling frequency of 1 kHz before PCI and 24 h post-PCI. The study obtained full approval from the Institutional Review Board of Shandong University (#2020S347) and was conducted according to the principles in the Declaration of Helsinki and its following amendments. Written informed consent was obtained from all participants before participation.

### 2.3. Data Preprocessing

R peaks were detected and ectopic R peaks were deleted based on a template matching approach [24] followed by visual inspections by experts. The RR interval time-series were formed by intervals of consecutive normal R waves. Figure 1 shows examples of RR interval time-seriesfrom the same CAD patient before and after PCI.

### 2.4. Analysis of HRV Time Series

For an RR interval time-series $\left\{RR_{1},RR_{2},\ldots RR_{i},RR_{i+1},\ldots ,RR_{N},1\leq i\leq N\right\}$ that consists of *N* RR intervals, time- and frequency-domain analysis methods classical entropy measures—sample entropy (SampEn) [17], distribution entropy (DistEn) [11], conditional entropy (CE) [19] and asymmetric analysis—Porta’s index (PI) [12], Guzik’s index (GI) [13], slope index (SI) [14] and area index (AI) [15] were analyzed.

#### 2.4.1. Time Domain Analysis

Standard deviation of the normal-normal intervals (SDNN):

(1)$$SDNN=\sqrt{\frac{\sum_{i=1}^{N}{\left(RR_{i}-RR_{mean}\right)}^{2}}{N}}$$

the root mean square of successive differences (RMSSD):

(2)$$RMSSD=\sqrt{\frac{\sum_{i=1}^{N-1}{\left(RR_{i+1}-RR_{i}\right)}^{2}}{N}}$$

#### 2.4.2. Frequency Domain Analysis

If the frequency range of RR intervals is 0.04–0.15 Hz the spectral component is considered as low frequency (LF) and 0.15–0.4 Hz is considered as high frequency (HF) [28,29]. LF/HF means the ratio of LF and HF.

#### 2.4.3. Entropy Analysis


Sample entropy (SampEn)SampEn is a measure of complexity which does not include self-similar [17]. For a time-series $X=\{x_{1},x_{2},\ldots x_{i},x_{i+1},\ldots ,x_{N-1},x_{N}\}$, given embedding dimension *m* and tolerance *r*.Form (*N − m*) vectors:(3)$$u_{i}^{\left(m\right)}=\{x_{i},x_{i+1},\ldots ,x_{i+m-1}\},(1\leq i\leq N-m)$$Then the SampEn is defined as: (4)SampEn=−log(A/B)
wherein, *A* and *B* are the number of template vector pairs having $D\left[u_{m+1}(i),u_{m+1}(j)\right]<r,i\neq j$ of length *m +* 1 and $D\left[u_{m}(i),u_{m}(j)\right]<r,i\neq j$ of length *m*, respectively; *D* is the Chebyshev distance.Distribution Entropy (DistEn)DistEn takes full advantage of the complete information by estimating the Shannon entropy of all distances [11]. The empirical probability density function of the distance matrix $d_{i,j}\leq r$ except the main diagonal (i.e., i≠j) is estimated by a histogram approach with a fixed bin number *B.* The probability of each bin is denoted as $\left\{p_{t},t=1,2,\ldots ,B\right\}$. DistEn can be defined by the following formula:(5)$$DistEn(m,\tau ,B)=-\frac{1}{\log_{2}(B)}\sum_{t=1}^{B}p_{t}\log_{2}(p_{t})$$
wherein, *m* is the dimension, *τ* is the time delay.Conditional Entropy (CE)Conditional entropy (CE) calculates the information contained in the new sampling point given the previous point [19,30]. Set a fix number of ξ (here it is the quantization level) values labelled 0~ξ−1 to coarse-grain X. It renders x(i) sequences of symbols $\hat{x}(i),i=1,2,\ldots ,N$. Form $u_{m}(i)$ and $u_{m+1}(j)$ by:(6)$$u_{m}\left(i\right)=\left[\hat{x}\left(i\right),\hat{x}\left(i-\tau \right),\ldots ,\hat{x}\left(i-\left(m-1\right)\tau \right)\right]$$
(7)$$u_{m+1}\left(j\right)=\left[\hat{x}\left(j\right),u_{m}(j-\tau )\right]$$
where (m−1)τ+1≤i,j≤N. The vectors $u_{m}(i)$ and $u_{m+1}(j)$ can be rewrote in the following format:(8)$${\left\{u_{m}\left(i\right)\right\}}_{10}=\hat{x}\left(i\right)\xi^{m-1}+\hat{x}\left(i-\tau \right)\xi^{m-2}+\ldots +\hat{x}\left(i-\left(m-1\right)\tau \right)\xi^{0}=w_{i}$$
(9)$${\left\{u_{m+1}\left(j\right)\right\}}_{10}=\hat{x}\left(j\right)\xi^{m}+{\left\{u_{m}\left(j-\tau \right)\right\}}_{10}=z_{j}$$The range of e is $0\sim \left(\xi -1\right)\sum_{i=1}^{m-1}\xi^{i}$, and the range of $z_{j}$ is $0\sim \left(\xi -1\right)\sum_{j=1}^{m}\xi^{j}$. Then CE can is defined by,
(10)$$CE(m,\tau )=SE(z_{j})-SE(w_{i})+perc(m)SE(1),$$
where SE(⋅) represents the percentage patterns $w_{i}$ found only once in the data set, the Shannon entropy of the quantized series $\hat{u}\left(i\right)$.


#### 2.4.4. Asymmetry Analysis


Porta’s indexPI was defined as the quotient by dividing the number of points below LI by the total number of points in Poincaré plot except those that are located on LI [12]. The way to calculate PI is:(11)$$PI=\frac{b}{m}\times 100$$
wherein, b means the number of points below LI, and m means the number of points not on LI.Guzik’s index (GI)GI measures the distance asymmetry of HRV series in Poincaré plot [13], which can be calculated by:(12)$$GI=\frac{\sum_{i=1}^{l}D_{i}}{\sum_{i=1}^{m}D_{i}}\times 100$$
wherein, l means the number of point above LI; m means the number of points in Poincaré plot except those that are on LI; $D_{i}$ is the distance of point $P_{i}$ to LI that can be calculated as:(13)$$D_{i}=\frac{\left|RR_{i+1}-RR_{i}\right|}{\sqrt{2}}$$Slope index (SI)Karmakar et al. proposed SI to measure the phase angle asymmetry of HRV series in Poincaré plot [14]. SI can be calculated by:(14)$$SI=\frac{\sum_{i=1}^{l}\left|R\theta_{i}\right|}{\sum_{i=1}^{m}\left|R\theta_{i}\right|}\times 100$$
wherein, l means the number of points above LI, and m means the number of points not on LI. $R\theta_{i}=\theta_{LI}-\theta_{i}$, where $\theta_{LI}$ is the phase angle of LI, and $\theta_{i}$ is the phase angle of every point which is defined as $\theta_{i}=atan\left(RR_{i+1}/RR_{i}\right)$Area index (AI)AI is a metric to measure the asymmetry of HRV series by using two dimensions of distance and angle [15]. AI can be calculated by the following formula:(15)$$AI=\frac{\sum_{i=1}^{l}\left|S_{i}\right|}{\sum_{i=1}^{m}\left|S_{i}\right|}\times 100$$
wherein, *l* and *m* represent the number of points above LI and total number of points in the Poincaré plot not on LI, respectively. *S_i_* is the area of the *i*th sector which can be calculated by $S_{i}=\frac{1}{2}\times R\theta_{i}\times r^{2}$, wherein, *r* is the radius of the sector.


#### 2.4.5. Parameters Selection

For SampEn, the parameters are set *m* = 2 and *r* = 0.2*σ*, wherein *σ* was the standard deviation of each realization. The parameters are set *m* = 2, *τ* = 1 and *B* = 256 for DistEn, and set *m* = 2, *τ* = 1 and *ξ*= 6 for CE. For SI, use the minimum of RR intervals as the reference point, and for AI, set the mean of RR intervals as the reference point and the nearer LI as the reference line [31]. Besides, the values of HRA equals to 50 means symmetric, and heart rate is asymmetric whether it’s greater than 50 or less than 50. Hence, the values of HRA indices were obtained by subtracting 50 from the value of the original HRA indices and taken absolute values [31]:(16)$$\begin{array}{l} Asymmetry\;of\;PI=\left|PI-50\right|,\\ Asymmetry\;of\;GI=\left|GI-50\right|,\\ Asymmetry\;of\;SI=\left|SI-50\right|,\\ Asymmetry\;of\;AI=\left|AI-50\right| \end{array}$$

### 2.5. Statistical Analyses

All results were first subjected to an examination of normality using the Shapiro-Wilk test. The non-parametric Wilcoxon signed-rank tests would be used to examine the difference before and after PCI if normality was rejected; otherwise the paired-*t* tests would be used. As a secondary analysis, we also performed a linear mixed-effect model to explore the differences in the 12 HRV measures across male and female with adjustment of age. The Bonferroni correction was used to correct multiple comparisons, thus the statistical significance was accepted at alpha level of 0.05/12 ≈ 0.0042. An effect size measured by the Cohen’s *d* static was also reported. The effect size was considered small if *d* < 0.5. All statistical analyses were performed using the Matlab software (Ver. R2016b, The MathWorks Inc., Natick, MA, USA).

## 3. Results

Figure 2 summarizes all indices results for CAD patients before and after PCI procedure. The effect size of all indices of CAD patients before and after PCI are showed in Figure 3. The *p* values of DistEn and CE were obtained by paired-*t* tests as they were normal distribution before and after PCI that were shown in Figure 2g,h. PI and AI were normal distribution before PCI while they were non-normally distribution after PCI, and the rest indices performed non-normally distribution both before and after PCI. The Wilcoxon signed-rank tests were used to decide whether the measures performed statistical significance difference between before and after surgery (Figure 2a–f,i–l). The SDNN, RMSSD, LF, HF, CE, PI, GI and AI exhibited certain reductions after surgery, whereas LF/HF, SampEn, DistEn and SI showed an opposite change. However, no statistical significance was indicated between CAD patients before and after PCI procedure for SDNN, RMSSD, LF, HF, LF/HF, SampEn, DistEn, GI and SI (*p* > 0.36), whereas there was a significant reduction in PI (Figure 2i, *p* < 0.015, Figure 3, *d* = 0.70), AI (Figure 2l, *p* < 0.001, Figure 3, *d* = 0.88). After Bonferroni correction PI showed no significant difference (*p* > 0.042).

HRV results of male and female CAD patients assessed by all indices are summarized in Figure 4. Linear mixed model was used to analyze the effect of PCI and gender on HRV indices. Figure 5 shows the increased and decreased number of male and female CAD patients after PCI compared with before. Considering the variation trends of all patients, relatively consistent changes were observed for female patients by the heart rate asymmetry indices, while random changes for male patients by SDNN GI and SI. Random changes for all patients were captured by SDNN, RMSSD, LF, HF, LF/HF, DistEn and CE, whereas the male increasing trend was dominant for SampEn. Consistent decreasing of both male and female patients was observed by AI. However, all indices indicated sex has no significant effect on HRV before and after PCI surgery (*p* > 0.07).

## 4. Discussion

In this pilot study, changes of HRV in twenty-seven coronary artery disease (CAD) patients before and after the PCI procedure were investigated using traditional time- (i.e., SDNN and RMSSD) and frequency-domain measures (i.e., LF, HF, and LF/HF), Poincaré plot-based HRA metrics (i.e., PI, GI, SI, AI), and entropy measures for nonlinear complexity/irregularity (i.e., SampEn, DistEn, and CE). Results demonstrate that all measures showed a relatively consistent trend. However, only the change in PI (*p* = 0.015) and AI (*p* < 0.002) were statistically significant.

### 4.1. Effect of PCI on Time- and Frequency-Domain Analysis Methods in Patients with CAD

Time-domain indices showed no significant difference between the two groups (*p* > 0.36, *d* < 0.09 for both SDNN and RMSSD). The high frequency (HF) component is controlled by sympathetic [28], and the low frequency (LF) component is sensitive to the activity of sympathetic and parasympathetic [32]. LF/HF indicates the degree of balance between sympathetic and parasympathetic activities [4]. No statistically significant difference (*p* > 0.32, *d* < 0.23) in all mentioned frequency-domain indices were observed between patients with CAD before PCI and post-PCI. It was speculated that with the improvement of myocardial ischemia, the improvement of autonomic nervous system (ANS) might be too weak to be captured by these traditional approaches.

### 4.2. Effect of PCI on Complexity in Patients with CAD

Sample entropy (SampEn), as a classical nonlinear metric, has been widely used in measuring the complexity of physiological signals [17]. The SampEn suggested that the complexity of HRV had no statistically significant change after PCI procedure (*p* > 0.79, *d* = 0.05). Additionally, maybe the complexity of HRV needs a long time (more than 24 h after the surgery) to increase significantly. Distribution entropy (DistEn), which quantifies the complete information by estimating the Shannon entropy of all distances [11], also showed no significant changed complexity after PCI (*p* = 0.49, *d* = 0.1). Conditional entropy (CE) evaluate the information contained in the new sampling point compared to the previous one, which showed the statistical results were *p* = 0.72, *d* = 0.07. Previous studies showed that nonlinear dynamic characteristics of RR intervals are significantly lower in patients with CAD than in healthy subjects [33,34]. The entropy measures of RR intervals after operation does not show significant change compared with that before surgery. That is, the cardiovascular system cannot return to the healthy state when in 24 h after PCI surgery. We guessed the autonomic nervous system takes a certain amount of time to recover.

### 4.3. Effect of PCI on Heart Rate Asymmetry in Patients with CAD

Heart rate asymmetry (HRA), i.e., the acceleration and deceleration of the heart rate, is a method for measuring autonomic nervous system function based on the asymmetric distribution of the RR interval series in the Poincaré plot [13]. The activation of the sympathetic nervous system speeds up the heartbeat, and the parasympathetic nervous system slows down the heartbeat [32,35]. PI (*p* = 0.02, *d* = 0.70) assesses the number asymmetry, GI estimates the HRA by distance and SI measures the HRA only by phase angles, which cannot catch the slight change in ANS of CAD patients after PCI within 24 h (*p’s* > 0.01, *d’s* < 0.7). AIsignificantly reduced in CAD patients after PCI (*p* = 0.002, *d* = 0.88) which supported that AI combined information from two aspects (distance and phase angle) could get more information hidden in the short-term RR intervals. This is consistent with previous studies that AI has better stability and consistency for short-term time series and cardiovascular disease causes AI to increase [15,31]. The values of HRA indices were increased in pathological situations [14,36,37], while researchers found the lower irreversibility measures mean a loss of complexity and pathological states [38,39]. Time irreversibility means that the statistical characteristics of the signal are not invariant when time reversal, while asymmetry refers to the distribution of the signal is imbalanced and/or disproportionate [40]. Irreversibility and asymmetry are not exactly the same, but there is also a certain correlation. HRA, an asymmetry method, measures the acceleration or deceleration of heart rate to quantify the activity of ANS from different angles. The way to describe the distribution of points in Poincaré plot may be one of the reasons for this phenomenon. However, the mechanism of HRA needs more studies in future.

### 4.4. Effect of Gender and PCI on HRV Indices in Patients with CAD 

AI of almost all patients showed an increased trend. The other three HRA indices performed relatively consistent decrease (PI) or increase (GI and SI) changes for the 10 female patients, while for male, they performed random increase or decrease (Figure 4 and Figure 5). The results of linear mixed model analysis indicated that the change of all indices was independent of sex.

Noticing that the HRA of female patients increases or decreases consistently. But the mechanism of gender impact on asymmetry of RR intervals is still unclear. In future, we are planning to investigate systematically the influence of gender in HRA, especially in HRA of the recovery of CAD patients after PCI procedure. What needs attention is that it is an algorithm without parameters, though not all of the patients have the same change direction. We suggested that the potential of AI in clinical application should be properly considered.

### 4.5. Study Limitation

One limitation of the study is the small sample size that may have limited our statistical power to detect the differences in some of the measures. A second limitation is the lack of long-term follow up on clinical outcomes in these participants. Follow-up studies with bigger sample sizes are warranted to verify the observations in this pilot study and to further examine the potential of these novel HRV analytical approaches to be used as a functional assessment tool for PCI outcome, as well as a way to enable long-term ambulatory monitoring of patients after PCI procedure. 

## 5. Conclusions

PCI procedures can immediately achieve revascularization of stenotic coronary arteries and rapidly improve myocardial ischemia in CAD patients that may result in a tiny improvement of ANS. Through the investigation of this pilot paper, it is found that by analyzing 27 samples, PI and AI can statistically capture this weak change. Besides, for male and female patients, AI obviously showed the same trends, so sex does not affect the their effect in monitoring cardiovascular function status. It is indicated that AI should be considered as a potential reference indicator for monitoring the recovery of cardiovascular system function in patients after PCI. 

## Figures and Tables

**Figure 1 entropy-23-00540-f001:**
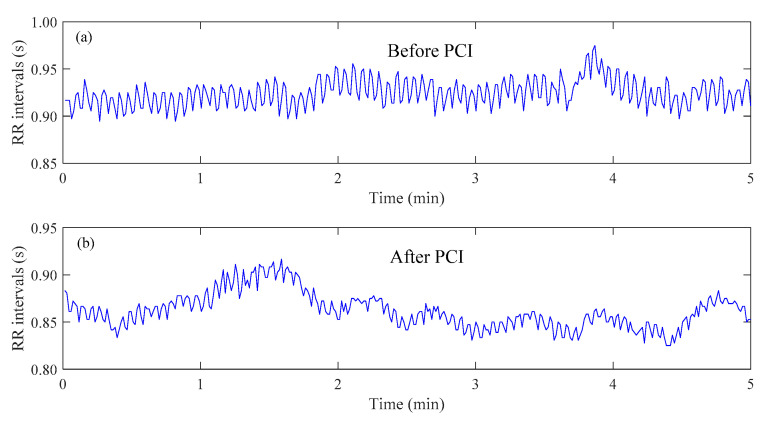
Examples of heartbeat interval (RR interval) time-series. (**a**) signal from a representative coronary artery disease (CAD) patient before percutaneous coronary intervention (PCI); (**b**) signal from the same CAD patient after PCI.

**Figure 2 entropy-23-00540-f002:**
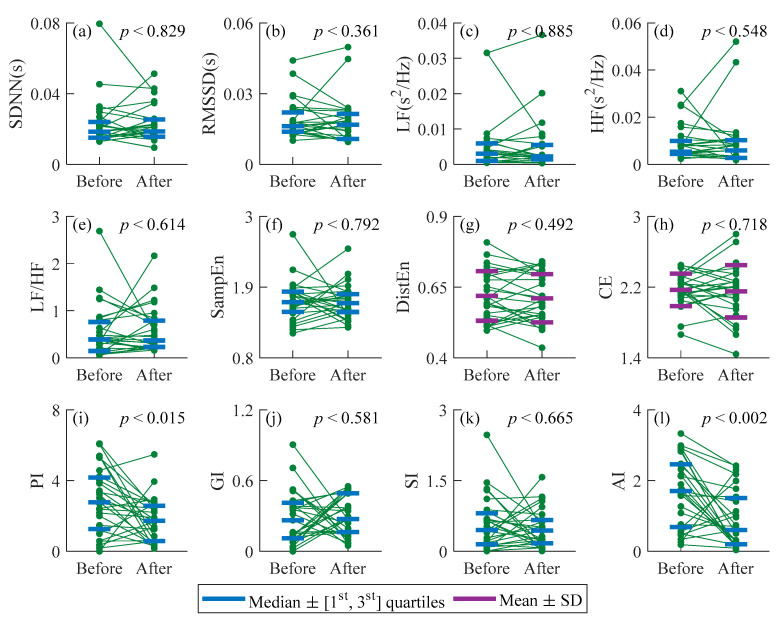
Summarized indices results for CAD patients before and after PCI procedure. In order to show the changes, results from the same individual were connected by lines. (**a**) SDNN; (**b**) RMSSD; (**c**) LF; (**d**) HF; (**e**) LF/HF; (**f**) SampEn; (**g**) DistEn; (**h**) CE; (**i**) PI; (**j**) GI; (**k**) SI; (**l**) AI. Before: before PCI procedure; After: after PCI procedure.

**Figure 3 entropy-23-00540-f003:**
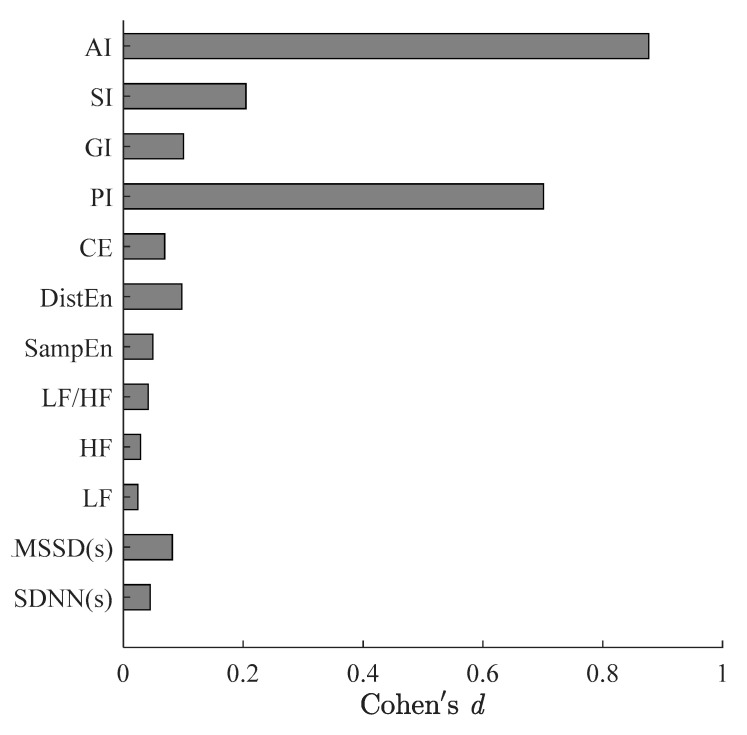
Effect size of all indices of CAD patients before and after PCI.

**Figure 4 entropy-23-00540-f004:**
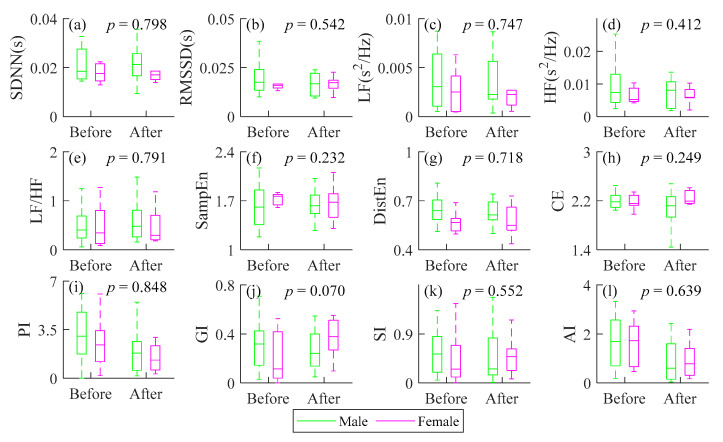
The results of short-term heart rate variability for male and female CAD patients before and after PCI procedure, respectively. (**a**) SDNN; (**b**) RMSSD; (**c**) LF; (**d**) HF; (**e**) LF/HF; (**f**) SampEn; (**g**) DistEn; (**h**) CE; (**i**) PI; (**j**) GI; (**k**)SI; (**l**) AI. *p*: the difference in each metrics across gender with adjustment of age. explored by linear mixed-effect model; Before: before PCI procedure; After: after PCI procedure. Results from males are marked in green and from females are marked in magenta.

**Figure 5 entropy-23-00540-f005:**
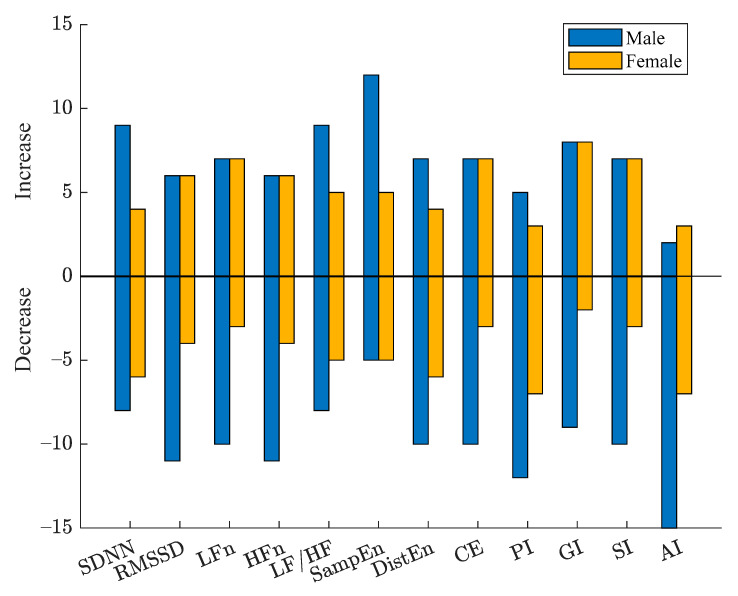
The increased (positive value) and decreased (negative value) number of male and female CAD patients after PCI compared with before.

**Table 1 entropy-23-00540-t001:** Basic characteristics of CAD patients.

Variables	Value
No. (male/female)	27 (17/10)
Age (years)	60.9 ± 9.9
Height (cm)	166.4 ± 8.9
Weight (kg)	71.4 ± 12.4
BMI (kg/m^2^)	25.6 ± 3.2
HR (No./min)	67.9 ± 11.3
SBP	131.3 ± 16.9
DBP	80.8 ± 10.0

Note: value is expressed as number or mean ± standard deviation (SD). BMI: body mass index, HR: heart rate, SBP: systolic blood pressure, DBP: diastolic blood pressure. All the indices were measured before surgery.

## Data Availability

The data presented in this study are available on reasonable request from the corresponding author, C.L. The data are not publicly available due to their containing information that could compromise the privacy of research participants.

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
