# Peer review of "Short-Term Effect of Percutaneous Coronary Intervention on Heart Rate Variability in Patients with Coronary Artery Disease"

_entropy, 2021, doi:10.3390/e23050540_

Round 1
Reviewer 1 Report
I read the paper by Chang Yan et al with pleasure. This is a well designed and well executed study, with careful and clear analyses. The authors carefully follow the methodology of a publishable study.
The only reservation one could have is a small sample size. I don't believe that it is sufficient to reach statistical significance with the use of this group. The authors are aware of this and discuss it in the limitations of the study, however, I would recommend that they slightly weaken the conclusions by stating the exploratory nature of the study, rather than providing a categorical statistical statements.
Another problem I noticed is that the Authors say that HRV measures are non-linear. In fact, most of the commonly used ones, including some of the ones used in the reviewed paper, are linear - see for example the works of Brennan, e.g. "Do existing measures of Poincaré plot geometry reflect nonlinear features of heart rate variability?". I think the Authors could make use of the considerations in this paper.
Another thing is the language - English does not seem to be the Authors' first language and there are really many small errors or unnatural sounding statements, especially towards the end of the paper. I would recommend that the authors try and find a native speaker to help them in this respect.
All in all this is a good paper and in my opinion it should be published in Entropy.
Reviewer 2 Report
Yan et al. examined the effect of percutaneous coronary intervention on the heart rate variability in 27 patients. They found that significant change in only AI and PI, not other HRV parameters. They concluded that PCI induces significant changes in some Entropy parameters. The short-term RR interval measurement is a very useful step forward towards an assessment of the effect of PCI. Nevertheless, I have a rather skeptical perspective on a fundamental level referring to the added value the paper provided from a scientific and a practical point of view.
Authors need to do the major revision before the paper is considered for publication. Details are given below:
1) Introduction needs to be further improved adding the state-of-the-art literature.
2) Several of the Entropy measures are already known thing and hence authors should only report if there is any novelty. Authors need to explain rationale for choosing Entropy measures for this research.
3) Number of patients was too small to conclude the effect of PCI on the Entropy measures.
4) More detailed clinical information is required including the prevalence of diabetes, chronic kidney disease, heart failure, history of stroke, peripheral arterial disease, myocardial infarction, left ventricular ejection, number of culprit, operation procedures, and medications. Also, please explain the reason that you excluded patients with left ventricular ejection fraction < 50%.
Reviewer 3 Report
The study assesses the effect of percutaneous coronary intervention (PCI) on cardiac control as evaluated from heart rate variability (HRV) indexes in patients with coronary artery disease (CAD).
The study is interesting but interpretation of the results should be deepened and methodological clarifications are needed.
- It is unclear the rationale underlying the selection of HRV markers. A more complete set of HRV indexes can be found in Maestri R et al, J Cardiovasc Electrophysiol, 18: 425-433, 2007. The limited set of markers utilized in this study must be defended. In absence of a strong rationale the set must be enlarged.
- The set of conditional entropy-based markers of complexity utilized in this study is extremely limited. Please see Porta A et al, IEEE Trans Biomed Eng, 64, 1287-1296, 2017 for a larger set of conditional entropy-based indexes frequently exploited in HRV study.
- The most robust HRV marker of vagal modulation, namely the power of HRV in the high frequency band expressed in absolute unit (see Hirsch JA and Bishop B, Am J Physiol 241: H620-H629, 1981) is missing and must be added.
- The decrease of irreversibility markers is usually taken as a negative finding indicating a loss of complexity and pathological situations (see Porta A et al, Phil Trans R Soc A, 367, 1215-1218, 2009 and De Maria B et al, PLoS ONE, 16, e0247145, 2021). Surprisingly, indexes of asymmetry decreased after PCI. This finding must be more deeply discussed and interpreted.
- The sum of LFnu and HFnu is 100. Given the full correlation between the two markers, one must be deleted.
- Please check the scale of panels in Fig.2. For example, PI and GI are bounded between 0 and 100 and typical values are slightly above 50. The same range is relevant to LFn and HFn indexes with typical values very different from those reported in the panels.
- CAD population is frequently studied with the aim at assessing cardiac neural control and baroreflex regulation (see Sandrone G et al, Ann Noninvasive Electrocardiol, 3, 237-243, 1998; Lucini D et al, Am Heart J, 143, 977-983, 2002; Neves VR et al, Clin Auton Res, 22, 175-183, 2012; de Oliveira Gois M et al, Med Biol Eng Comput, 57, 1405-1415, 2019). Therefore, the originality of the study might appear to be limited. Discussion should be enlarged to account for differences with previous studies, thus emphasizing the novelty of the present piece of work.
- Please clarify the emphasis on AI (see e.g. the abstract). Also Pi was able to separate post-condition from pre-condition.
- Please check definition of the acronyms (e.g. in the abstract).
Round 2
Reviewer 2 Report
The authors have provided acceptable detailed responses to reviewers' critiques and have appropriately revised the manuscript.
Reviewer 3 Report
The manuscript was improved. The authors replied satisfactorily to all my issues and followed carefully the suggestions given.